# PCPs: Patient Cardiac Prototypes to Probe AI-based Medical Diagnoses, Distill Datasets, and Retrieve Patients

**Dani Kiyasseh**                                    *danikiy@hotmail.com*
*Department of Computing and Mathematical Sciences*
*California Institute of Technology*
*Pasadena, CA, USA*

**Tingting Zhu, David Clifton**                      *first.last@eng.ox.ac.uk*
*Department of Engineering Science*
*University of Oxford*
*Oxford, UK*

**Reviewed on OpenReview:** *https://openreview.net/forum?id=X1pjWMCMBO*

## Abstract

Clinical deep learning systems often generate population-based and opaque medical diagnoses. This is in contrast to how primary care physicians make decisions, often adapting population-based protocols to the unique patient under consideration. Inspired by the workflow of such physicians, we develop a framework for learning embeddings, referred to as patient cardiac prototypes (PCPs), which capture information that is unique to an individual patient's electrocardiogram (ECG) data. Through rigorous evaluation on three publicly-available ECG datasets, we show that PCPs allow researchers to inspect why a particular diagnosis was made. We also demonstrate that PCPs are effective dataset distillers, where they can be used to train a model in lieu of a dataset orders of magnitude larger to achieve comparable performance. We show that PCPs can also be exploited to retrieve similar patient data across clinical databases. Our framework contributes to the development of transparent and patient-specific clinical deep learning systems.

## 1 Introduction

Primary care physicians often account for the complex mosaic of a patient's demographics and physiological state by adapting population-based protocols to the unique patient under their care (Hamburg & Collins, 2010). Such a patient-specific approach is in contrast to, for example, randomized controlled trials, long considered the gold standard of evidence in medical research (Cartwright, 2007), where findings are population-based and may overlook patient-specific nuances (Akobeng, 2005). These characteristics are all too common in deep learning systems which, despite their success in automating the diagnosis of medical conditions such as cardiovascular diseases (Galloway et al., 2019; Attia et al., 2019a;b; Ko et al., 2020), continue to generate population-based and opaque diagnoses.

We believe that, akin to primary care physicians, clinical deep learning systems can benefit from explicitly incorporating information about an individual patient's data into their decision making. For example, the explicit dependence of deep learning-based predictions on unique patient data can enable researchers to inspect *why* such predictions were made. Operating at the patient level also allows for the retrieval of patient data, where a clinical database is searched in order to retrieve data that are most similar to the data of an existing patient. Such retrieval capabilities can allow physicians to compare the disease and treatment trajectories of similar-looking patients, and enable medical educators to leverage up-to-date real-world evidence as a means of teaching the next generation of medical students.

Previous studies have attempted to capture information that is unique to an individual patient's data. This has primarily been achieved through graph neural networks (Lu & Uddin, 2021; Hernández-Lorenzo et al., 2022) where each node in the graph reflects a unique patient and the edges reflect the similarity of patients (Sharafoddini et al., 2017; Pai & Bader, 2018; Pai et al., 2019). These approaches, however, assume that an individual patient's data are already succinctly represented by a set of features and that the similarity between patients are known a priori, often reflected through an adjacency matrix. We instead learn these compact patient representations from scratch which can subsequently be used, for example, to discover the adjacency matrix. Driven by the recent success of contrastive learning (Cheng et al., 2020), researchers have focused on learning patient-specific representations of cardiac signals as a pre-training step (Kiyasseh et al., 2021b) or on learning prototypes of patient *groups*, such as those sharing the same sex, age, and disease class attributes (Kiyasseh et al., 2021a). Prototypes that have been introduced in the past are either exclusively limited to the few-shot learning application (Snell et al., 2017; Sung et al., 2018) or to clustering data points in an unsupervised manner (Li et al., 2021). In contrast, we develop a *supervised* contrastive learning framework for the learning of prototypes with multiple use-cases.

Our main contributions are fourfold. First, to explicitly incorporate information about a patient's data into a deep learning system, we develop a framework that learns embeddings which we refer to as patient cardiac prototypes (PCPs) via supervised contrastive learning (§2.2). By design, PCPs capture information unique to an individual patient's electrocardiogram (ECG) data (§4.2). Through rigorous evaluation on three publicly-available ECG datasets, we demonstrate that PCPs allow for the probing of deep learning-based predictions, enabling researchers to inspect *why* a particular prediction is made (§4.4). This can contribute to the development of *transparent* clinical deep learning systems and instill trust in clinical stakeholders. We also show that PCPs are effective dataset distillers, where they can be used to train a model and match the performance of one trained on the full dataset (§4.6). In the process, we demonstrate that PCPs outperform state-of-the-art core-set construction methods. This capability the potential to reduce the amount of resources required to train deep learning systems. Lastly, we show that PCPs can reliably retrieve similar (and dissimilar) patient data across distinct clinical databases (§4.7), thereby potentially facilitating the provision of education by medical educators to the next generation of medical students.

## 2 Patient cardiac prototypes: motivation and design

### 2.1 Leveraging spatial and temporal invariances

Recent research has demonstrated the benefit of learning representations of cardiac signals that are invariant to spatial and temporal sources of variability while *pre-training* on large-scale cardiac datasets (Kiyasseh et al., 2021b). An invariance is an aspect of the input data which, when changed, does *not* alter the information exhibited by that data. That study demonstrated that attracting representations of different ECG leads to one another or attracting representations of temporally-adjacent ECG signals to one another (e.g., on the order of seconds) can allow deep learning systems to perform cardiac arrhythmia classification with fewer training data points than would otherwise have required. Inspired by those findings, we decided, in this study, to leverage the same spatial and temporal invariances to learn representations (PCPs). We assume that data that belong to the *same* patient reflect the *same* type of information, otherwise known as intra-patient invariance (see Appendix A for additional motivation).

### 2.2 Learning patient cardiac prototypes

**Notation** Let us assume we have a dataset, $\mathcal{D} = \{\boldsymbol{x_i}, y_i\}_{i=1}^{N}$, comprising $N$ instances, $\boldsymbol{x}$, and cardiac arrhythmia labels, $y$, corresponding to a total of $\Omega_{\text{train}}$ patients in the training set. Each patient is associated with $N/\Omega_{\text{train}} > 1$ instances. This could be due to the provision of multiple medical tests during the same hospital visit or several visits. We also define a learner, $f_\theta : \boldsymbol{x} \in \mathbb{R}^D \to \boldsymbol{h} \in \mathbb{R}^E$, parameterized by $\boldsymbol{\theta}$, that maps a $D$-dimensional instance, $\boldsymbol{x}$, to an $E$-dimensional representation, $\boldsymbol{h}$. We aim to learn a set of embeddings, each of which efficiently summarizes the cardiac state of a patient. To that end, we associate each patient in the training set with a unique, learnable embedding, $\boldsymbol{p} \in \mathbb{R}^E$, to form the set of embeddings,

$P = \{\boldsymbol{p}_j\}_{j=1}^{\Omega_{train}}$. Hereafter, we refer to such embeddings as patient cardiac prototypes (PCPs). We next explain how to learn PCPs.

**Contrastive learning** We use the contrastive learning framework which, in short, consists of a sequence of attractions and repulsions between representations of instances. The idea is to attract representations of instances of the same patient to the single PCP of that same patient, and to repel them from the PCPs of the remaining patients. Formally, we encourage the representation, $\boldsymbol{h}_i = f_\theta(\boldsymbol{x}_i)$, of an instance, $\boldsymbol{x}_i$, associated with the $k$-th patient to be similar to the $k$-th PCP, $\boldsymbol{p}_k$, and dissimilar from the remaining PCPs, $\boldsymbol{p}_j$, $j \neq k$ (see Fig. 1). To achieve this, we optimize a variant of the InfoNCE loss (Oord et al., 2018) (equation 1). Intuitively, it penalizes the learner for placing less probability mass on the similarity, $s(\boldsymbol{h}_i, \boldsymbol{p}_k)$, of the representation and prototype pair that should be most similar (based on patient ID) than on the similarity of other pairs, of which there are $\Omega_{train} - 1$. We quantify the cosine similarity between such pairs with a temperature parameter, $\tau$.

$$\mathcal{L}_{NCE} = -\sum_{i=1}^{B} \log \left[ \frac{e^{s(\boldsymbol{h}_i, \boldsymbol{p}_k)}}{\sum_{j}^{\Omega_{train}} e^{s(\boldsymbol{h}_i, \boldsymbol{p}_j)}} \right] \qquad s(\boldsymbol{h}_i, \boldsymbol{p}_j) = \frac{\boldsymbol{h}_i \cdot \boldsymbol{p}_j}{\|\boldsymbol{h}_i\| \|\boldsymbol{p}_j\|} \cdot \frac{1}{\tau} \tag{1}$$

As a result of this many-to-one mapping from representations to PCP, the latter will become invariant to *intra-patient* differences present in the data. For context, it is these prototypes, $P$, which are presented in Fig. 2 (left). This outcome is desirable only if we assume that such intra-patient differences point to the same underlying physiological state of the patient.

## 2.3 Incorporating patient cardiac prototypes into diagnosis pipeline

Equipped with PCPs, we direct our attention to the question of *how do we exploit PCPs to generate diagnoses?* Before addressing this question, we note that neural network parameters are often deterministic; they are held constant during inference. Therefore, when making a prediction (e.g., medical diagnosis), a network depends almost exclusively on the instance. Although an instance is likely to reflect patient information, a network does *not* explicitly exploit such information. In light of this, we design a framework, inspired by hypernetworks (Ha et al., 2016), in which a subset of the network parameters are explicitly conditioned on patient information.

**Medical diagnosis with hypernetworks** A hypernetwork, a neural network in and of itself, generates parameters for another neural network. Formally, a hypernetwork is a function, $g_\phi : \boldsymbol{h} \in \mathbb{R}^E \to \boldsymbol{\omega} \in \mathbb{R}^{E \times C}$, parameterized by $\boldsymbol{\phi}$, that maps an $E$-dimensional representation, $\boldsymbol{h}$, to a matrix of parameters, $\boldsymbol{\omega}$, where $C$ is the number of class labels (i.e., cardiac arrhythmia categories). The output parameters, $\boldsymbol{\omega}$, can now be used to parameterize a linear classification head, $p_\omega : \boldsymbol{h} \in \mathbb{R}^E \to \boldsymbol{y} \in \mathbb{R}^C$, which maps an $E$-dimensional representation, $\boldsymbol{h}$, to an output probability distribution, $\boldsymbol{y}$. To explicitly condition the parameters, $\boldsymbol{\omega}$, on patient information as we had initially desired, we exploit PCPs differently during the training and inference stages of the framework, as outlined next.

**Retrieving PCPs during training** During the training stage, each representation, $\boldsymbol{h_i} = f_\theta(\boldsymbol{x_i})$, of an instance, $\boldsymbol{x_i}$, serves multiple purposes (see Fig. 1 left). First, it is attracted to its corresponding PCP, $\boldsymbol{p_k}$, as outlined earlier. To do so, we optimize the InfoNCE loss. Second, it is used as an input to the hypernetwork to generate *instance-specific* parameters, $\boldsymbol{\omega_i} = g_\phi(\boldsymbol{h_i})$. Third, the representation is input into the classification head, $p_\omega$, as is usual with neural networks. Given the ground-truth disease class, $c$, of each instance in a mini-batch of size $B$, we can optimize the categorical cross-entropy loss ($\mathcal{L}_{CE}$). In summary, during the training stage, we learn the parameters of the feature extractor ($\boldsymbol{\theta}$), the hypernetwork ($\boldsymbol{\phi}$), and the PCPs ($\{\boldsymbol{p}_j\}_{j=1}^{\Omega_{train}}$), in an end-to-end manner by optimizing the combined loss ($\mathcal{L}_{combined}$).

$$\mathcal{L}_{CE} = -\sum_{i=1}^{B} \log p_{\omega_i}(\boldsymbol{y_i} = c | \boldsymbol{h_i}) \qquad \mathcal{L}_{combined} = \mathcal{L}_{CE} + \mathcal{L}_{NCE} \tag{2}$$

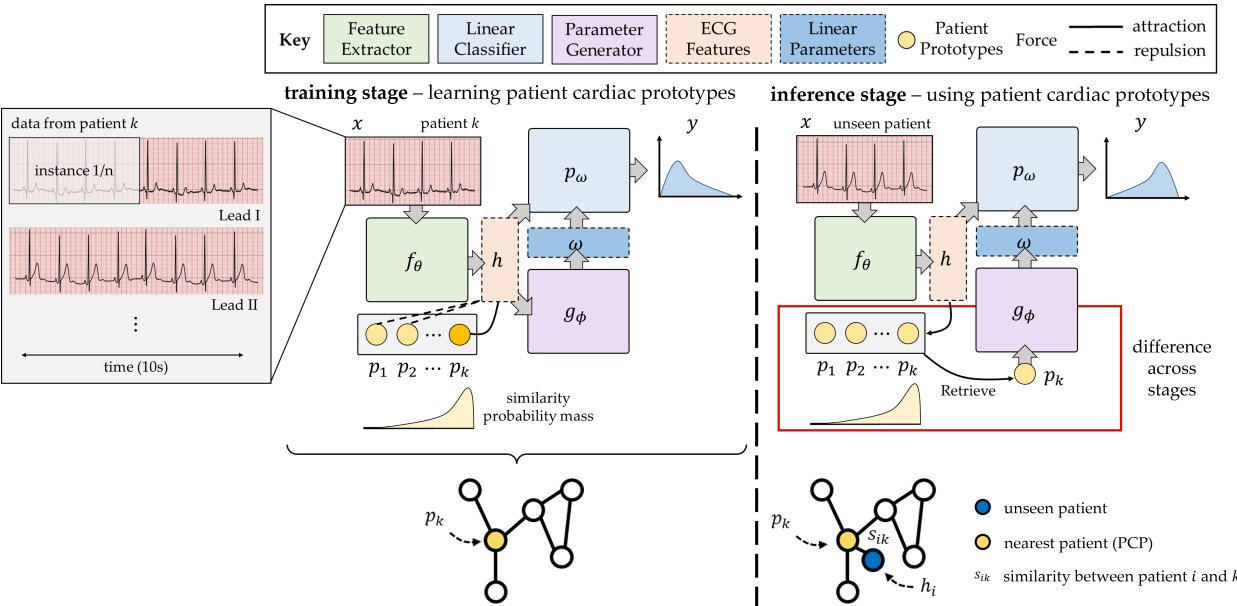

Figure 1: **Training and inference stages of cardiac arrhythmia diagnosis with patient cardiac prototypes. (training stage)** the representation, $\boldsymbol{h}$, of an instance, $x$, belonging to data from patient $k$ is (a) fed into a hypernetwork, $g_\phi$, to generate parameters, $\omega$, for a linear classification layer, $p_\omega$, (b) input directly into $p_\omega$ to output a class probability distribution, and (c) encouraged to be similar to the corresponding patient cardiac prototype (PCP), $\boldsymbol{p}_k$. This is akin to learning the features of a node in a graph from scratch. **(inference stage)** the representation of an instance associated with an unseen patient retrieves the nearest PCP, $\boldsymbol{p}_k$ (see graph analogy), which is then input into the hypernetwork. This generates linear parameters for classification.

**Retrieving PCPs during inference** During the inference stage, we propose several modifications to the pipeline (see Fig. 1 right). Since patients in the inference stage do *not* overlap with those in the training stage (by design), we no longer attract or repel each representation, $\boldsymbol{h_i}$, from PCPs. Instead, each representation searches through the set of PCPs, $P$, and retrieves the single PCP to which it is closest, $\boldsymbol{p}_k = \mathrm{argmax}_{\boldsymbol{p}_j}\ s(\boldsymbol{h_i}, \boldsymbol{p}_j)$, where $s$ is some similarity metric such as cosine similarity.

This similarity between patients is akin to the cross-attention mechanism used in Transformer architectures (Vaswani et al., 2017) or to edges connecting patients in a graph (Fig. 1, right). While it is possible to retrieve a single instance (as opposed to a PCP), doing so has several drawbacks. First, removing the PCP from the pipeline would make it difficult to reason at the patient level, which can be useful when probing the errors of a system (§4.4). Second, PCPs can conceptually be extended to multiple data modalities, thereby succinctly summarizing the clinical state of a patient more broadly. For the implications of multi-modal prototypes, please refer to the Discussion. Moreover, from a computational standpoint, searching through the full dataset to retrieve a single instance is more demanding than searching through the PCPs, which can be orders of magnitude smaller in size. This can introduce unwanted latency during inference. We nonetheless conducted such experiments and found a minimal change in the diagnostic performance of the system.

Once we retrieve the PCP, $\boldsymbol{p}_k$, it is used as an input to the hypernetwork, $g_\phi$. As a result, we generate parameters, $\boldsymbol{\omega_i} = g_\phi(\boldsymbol{p}_k)$, for each unseen instance in the held-out set of data. In other words, we generate parameters conditioned on the single PCP associated with the patient in the training set that is deemed most similar to the patient observed during inference. In summary, the final diagnosis is underpinned by this retrieval and parameter-generation process.

## 3  Experimental design

### 3.1  Electrocardiogram datasets

In this study, we experimented with three publicly-available datasets of electrocardiogram signals and cardiac arrhythmia labels. **CPSC** (Alday et al., 2020) consists of 12-lead ECG recordings from 6,877 patients alongside nine cardiac arrhythmia labels: AFIB, I-AVB, LBBB, Normal, PAC, PVC, RBBB, STD, and STE. **Chapman** (Zheng et al., 2020) consists of 12-lead ECG recordings from 10,646 patients alongside four high-level cardiac arrhythmia labels: AFIB, GSVT, Sinus Bradycardia, and Sinus Rhythm. **PTB-XL** (Wagner et al., 2020) consists of 12-lead ECG recordings from 18,885 patients alongside 71 different types of annotations provided by two cardiologists. We followed a previously-established training and evaluation protocol (Strodthoff et al., 2020) where we leveraged the 5 diagnostic class labels: Conduction Disturbance (CD), Hypertrophy (HYP), Myocardial Infarction (MI), Normal (NORM), and Ischemic ST-T Changes (STTC). We altered the original setup to only consider ECG segments with one label assigned to them and converted the task into a binary classification problem (NORM vs. Rest). Across all datasets, we split patients randomly into training, validation, and test sets, ensuring that there was no patient overlap between sets (see Appendix B).

### 3.2  Description of tasks and baselines

We aim to demonstrate the utility of patient cardiac prototypes in achieving three distinct tasks: cardiac arrhythmia classification, dataset distillation, and patient retrieval.

**Cardiac arrhythmia classification**  To achieve this task, we follow the previously-outlined diagnosis pipeline (Fig. 1). We also conduct an extensive set of ablation studies where we investigate the effect on performance of (a) retrieving a different number of patient cardiac prototypes during the inference stage (§4.3.1), (b) learning prototypes in a purely supervised manner ($\mathcal{L}_{CE}$ only) (§4.3.2), and (c) learning prototypes in a purely contrastive manner ($\mathcal{L}_{NCE}$ only) (§4.3.3). Additional details are provided in Appendix D.

**Dataset distillation**  In light of our interpretation of patient cardiac prototypes as efficient descriptors of the cardiac state of a patient, we leveraged them for the task of dataset distillation. Here, a compact and potentially synthetic dataset is used to train a model that matches the performance of one trained on the the full dataset, which is orders of magnitude larger (Wang et al., 2018; Cazenavette et al., 2022). In our context, we first learned the patient cardiac prototypes which were used to subsequently fit a machine learning model (e.g., support vector machine). We then evaluated the performance of this model on representations of unseen cardiac signals, as with the cardiac arrhythmia classification task described above. We benchmarked PCPs against state-of-the-art methods (§4.6) which construct compact subsets of data (core-sets) which we refer to as **Lucic** (Lucic et al., 2016), **Lightweight** (Bachem et al., 2018), and **Archetypal** (Mair & Brefeld, 2019).

**Patient retrieval**  By virtue of capturing information that is unique to a patient's cardiac data, we hypothesized that patient cardiac prototypes might also have the potential to search through an unseen corpus of data and retrieve patients whose cardiac attributes (e.g., disease class, heart rate, etc.) match those of the patient cardiac prototypes. To achieve this, we treat a patient cardiac prototype as a query and calculate its cosine similarity to all the representations of cardiac signals for patients in the unseen corpus. When aiming to retrieve *similar* patients, we inspect the pairs of patients with a higher cosine similarity (§4.7).

### 3.3  Evaluation metrics

For the tasks of **cardiac arrhythmia classification** and **dataset distillation**, we evaluated the performance of models by calculating the area under the receiver operating characteristic curve (AUC) of their predictions on unseen cardiac signals in a held-out set of data. For the task of **patient retrieval**, we quantify the proportion of data points retrieved which are in fact relevant. When aiming to retrieve *similar* patients, we define relevance as data points whose cardiac attributes match those of the query patient cardiac prototype. In contrast, when aiming to retrieve *dissimilar* patients, we define relevance as data points whose

attributes do *not* match those of the query prototype. These are equivalent to calculating the precision and negative predictive value (NPV) of the retrieved data points, respectively. Additional details can be found in Appendix E.

## 4 Experimental results

### 4.1 Patient cardiac prototypes are distinct and can distinguish between cardiac arrhythmias

To inspect the type of information captured by our learned patient cardiac prototypes, we visualized them via UMAP (McInnes et al., 2018) (see Fig. 2, left). We found that patient cardiac prototypes are distinct from one another. This is evident, qualitatively, by the lack of collapse of the PCPs to a select few points (Fig. 2, left), a favourable finding in our context. To see why, note that during inference on an unseen cardiac signal, prototypes are explicitly input into a hypernetwork that generates parameters enabling a cardiac arrhythmia classification (see Fig. 1 right, and Methods). Therefore, if prototypes were to collapse to a single point, then the same parameters will be generated regardless of the patient data input, diminishing their expressiveness. A more quantitative approach to supporting this claim is provided in the next section. We also found that PCPs do indeed reflect cardiac arrhythmia information. This is evident by the visible separability of the UMAP projections based on the cardiac arrhythmia categories.

### 4.2 Patient cardiac prototypes capture information unique to patient's ECG data

We explored the extent to which patient cardiac prototypes capture information unique to an individual patient's data. To do so quantitatively, we calculated the (Euclidean) distance between PCPs and representations of cardiac signals, belonging to either the same patient (*PCP to Same Training Patient*) or different patient (*PCP to Different Training Patient*). We present the distribution of these distance values for the Chapman dataset (Fig. 2, right).

We found that PCPs do indeed capture information that is unique to a patient's ECG data. This is evident by the smaller distance values of the *PCP to Same Training Patient* group (average = 4) than the *PCP to Different Training Patient* group (average = 9) (Fig. 2, right). This finding suggests that PCPs are twice as similar to representations of cardiac signals from the same patient than those from a different patient. Furthermore, for our proposed retrieval mechanism to work during the inference stage (see Fig. 1), the distance between representations of unseen cardiac signals and PCPs (Fig. 2, right, *PCP to Validation Patient*) must be reasonable and on the same order of magnitude. We found that these distance values

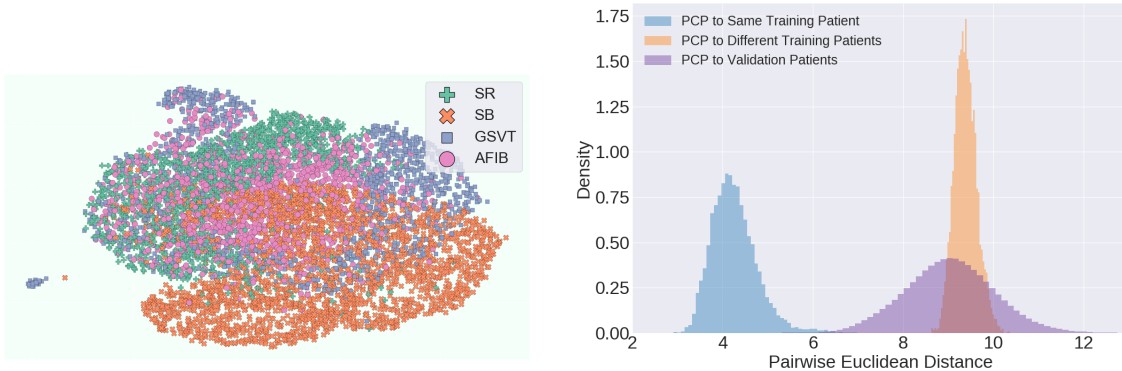

Figure 2: **Patient cardiac prototypes can distinguish between cardiac arrhythmias and capture information unique to individual patient's ECG data.** (left) UMAP projection of the PCPs with colours reflecting the four cardiac arrhythmia labels, Sinus Rhythm (SR), Sinus Bradycardia (SB), GSVT, and Atrial Fibrillation (AFIB). (right) Distance between PCPs and representations of instances in the training set associated with the same patient annotation, representations in the training set associated with a different patient annotation, and representations in the validation set.

are indeed reasonable since they are on the same order of magnitude as the distance values of the *PCP to Different Training Patients* group.

### 4.3 Ablation studies

#### 4.3.1 Effect of retrieval mechanism

During the inference stage, our framework is dependent on the retrieval mechanism (see Methods, Fig. 1). For example, to diagnose an unseen cardiac signal, we retrieve the *single* PCP that is closest to its representation. To appreciate this, note that the retrieval of an inappropriate PCP would have ramifications on the linear parameters that are generated by the hypernetwork, and in turn, the cardiac arrhythmia diagnosis. We therefore explored four variants of this retrieval mechanism which differ in the extent to which they leverage information unique to a patient's ECG data (see Appendix for details on variants, Fig. 3, left).

We found that incorporating imprecise information about an individual patient's ECG data hindered the generalization performance of the deep learning system. For example, the *mean* variant, in which all PCPs are simply averaged before retrieval, achieved AUC $\approx 0.65$ irrespective of the embedding dimension, which is significantly lower than that achieved by other variants of the retrieval mechanism. Incorporating some patient information resulted in a significant improvement in performance. For example, the *similarity-weighted mean* variant, in which we retrieve a weighted linear combination of PCPs, achieved AUC $\approx 0.75$ compared to 0.66 for *mean* at $E = 64$. This suggests that the PCP-derived similarity coefficients in the latter approach were beneficial, thus lending support to the utility of PCPs in the diagnosis pipeline. When limiting the retrieval mechanism to retrieve a *single* PCP (*nearest* variant), we found that individual PCPs were still relevant for diagnosis. For example, at $E = 64$, *similarity-weighted mean* and *nearest* achieved AUC $\approx 0.75$ and 0.89, respectively. Incorporating additional information from a subset of patients (*nearest 10* variant) further improved performance, albeit in a more marginal way.

#### 4.3.2 Effect of prototype definition

We also explored whether it was even necessary to incorporate prototypes into the deep learning system. To do so, we replaced an individual patient's prototype with the average representation of that patient (*mean representation*), without having to learn prototypes with a contrastive loss (Fig. 3, right). We found that mean representations capture information that is less unique to an individual patient's ECG data than do PCPs. This is evident by the smaller difference in performance across the variants of the retrieval mechanism when using mean representations relative to PCPs (Fig. 3, right). For example, the greatest difference in

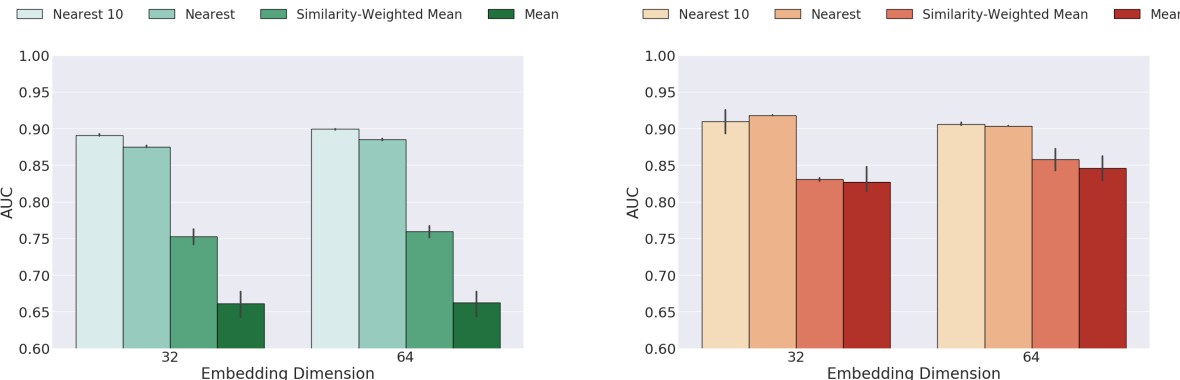

Figure 3: **Ablation studies examining the marginal impact of the components of our framework on performance.** Effect of the retrieval mechanism variants on the performance of the deep learning systems when using (left) PCPs and (right) embeddings learned without a contrastive loss, also referred to as *mean representations*.

the AUC $\approx 0.92 - 0.83 = 0.09$ and AUC $\approx 0.89 - 0.66 = 0.23$, for the mean representations and the PCPs, respectively.

### 4.3.3   Effect of supervision

To examine the importance of supervision on the learning of prototypes, we trained a variant of our framework without disease class labels, and subsequently used the prototypes for cardiac arrhythmia classification with $k$-nearest neighbours ($k$-NN). Here, $k$ reflects the number of prototypes retrieved during inference. We compare the performance of this approach (NCE Loss Only), which is akin to the work of Li et al. (2021), to that of patient cardiac prototypes (Combined Loss) (Appendix F, Fig. 8). As expected, we found that prototypes learned without cardiac arrhythmia label supervision perform more poorly than those learned with supervision.

### 4.4   Patient cardiac prototypes allow for the probing of deep learning-based diagnoses

The design of patient cardiac prototypes and their subsequent use in the diagnostic pipeline can allow machine learning practitioners to inspect *why* a deep learning-based medical diagnosis was made. Specifically, we hypothesized that correct deep learning-based predictions were more likely to be associated with a retrieved PCP whose patient characteristics (e.g., disease class, age, etc.) matched that of the unseen cardiac signal. The inverse would hold for an incorrect prediction. The intuition is that retrieving an irrelevant PCP, where relevance is based on matching patient characteristics, would lead to classification errors. We tested these hypotheses quantitatively and qualitatively.

Quantitatively, we inspected the proportion of correct (incorrect) deep learning-based predictions that were associated with a relevant (irrelevant) PCP retrieved during inference (Fig. 4, left). As expected, we found that while 94% of *correct* predictions were associated with the retrieval of a relevant PCP, 74% of *incorrect* predictions were associated with the retrieval of an irrelevant PCP. We reaffirm the latter qualitatively by randomly identifying an incorrect prediction and inspecting the retrieved PCP (Fig. 4, right). We found that the patient associated with retrieved PCP and that associated with the unseen cardiac signal exhibited different characteristics (disease class: supraventricular tachycardia vs. atrial fibrillation, age 63 vs. 87, ventricular rate: 164 vs. 91). These findings demonstrate how PCPs can be used to conduct an error analysis and shed light on why a misdiagnosis was made.

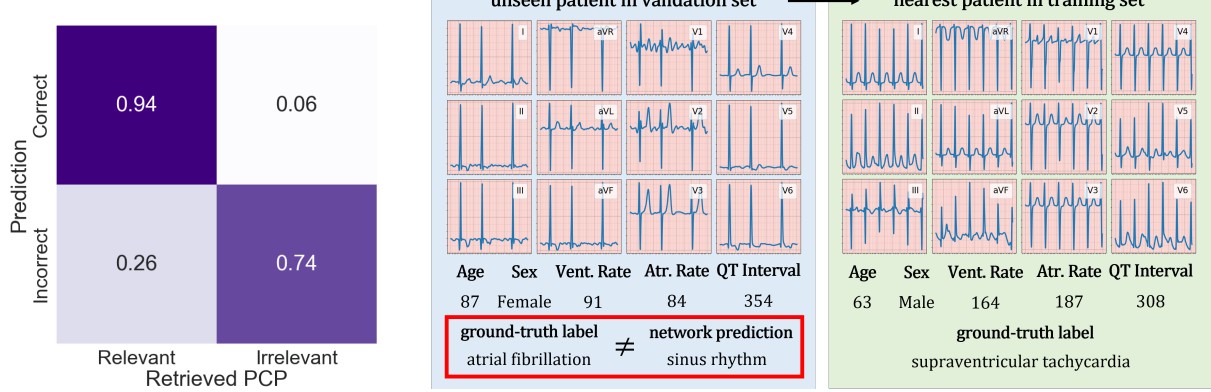

Figure 4: **Correctness of deep learning-based predictions often depends on relevance of retrieved PCP.** (left) Proportion of predictions in which the retrieved PCP is relevant to the unseen cardiac signal. (right) Incorrect prediction is associated with the retrieval of an irrelevant PCP. This approach facilitates error analysis and allows researchers to determine *why* an individual patient's diagnosis was made.

## 4.5 Patient cardiac prototypes outperform state-of-the-art dataset distillation methods

We *trained* a model exclusively on the learned patient cardiac prototypes and *evaluated* it on representations of unseen cardiac signals. We benchmarked patient cardiac prototypes against baseline methods when deployed on either raw cardiac signals or representations of such signals learned through our framework (Table 1).

We found that core-sets of raw instances, generated by baseline core-set construction methods, did not provide a sufficient training signal to allow machine learning models to achieve strong generalization performance. For example, on the Chapman dataset, Lucic, Lightweight, and Archetypal achieved AUC = 56.8, 56.6 and 54.8, respectively. We attribute this performance to the low class separability of the input features. However, the baseline methods continued to perform poorly even when provided with the opportunity to construct core-sets from representations learned via our framework. Recall that these representations are more separable along the disease class dimension (see Fig. 2 left and the associated performance in Fig. 3). For example, on the Chapman dataset, Lucic, Lightweight, and Archetypal achieved AUC = 57.8, 58.9 and 58.1, respectively. We found that PCPs outperformed these state-of-the-art core-set construction methods, instead achieving an AUC = 88.7.

## 4.6 Patient cardiac prototypes are effective dataset distillers

To determine whether PCPs are effective dataset distillers, we compared the performance of a model trained on 100% of the PCPs ($F = 1$) to one trained on the full training set. To explore the extent to which further distillation was possible, we randomly chose a fraction, $F \in [0.05, 0.1, 0.2, 0.5]$, of the PCPs and trained a machine learning model on that subset, while continuing to evaluate on the same held-out set of data (Fig. 5). We also depict the performance of these frameworks when trained on all instances in the larger, original dataset (horizontal, dashed lines), which is several folds larger than the number of PCPs.

We found that PCPs are indeed effective dataset distillers (Fig. 5). For example, when training on 100% of the PCPs ($\Omega_{train} = 6,387$), an SVM model achieved similar performance (AUC $\approx 0.89$) to one trained on all cardiac signals in the training set ($N = 76,614$). Expressed differently, similar performance was achieved despite a *12-fold* reduction in the number of training instances provided to the model. Such a finding suggests that the PCPs are able to capture the most pertinent information in the dataset and neglect

| Core-set | Chapman | CPSC | PTB-XL |
|---|---|---|---|
| *raw instances* | | | |
| Lucic | 56.8 (0.8) | 50.1 (0.1) | - |
| Lightweight | 56.6 (0.4) | 50.1 (0.1) | - |
| Archetypal | 54.8 (0.3) | 50.1 (0.1) | - |
| *representations* | | | |
| Lucic | 57.8 (17.5) | 50.6 (1.2) | 51.6 (4.5) |
| Lightweight | 58.9 (16.8) | 50.5 (1.2) | 52.4 (3.6) |
| Archetypal | 58.1 (16.8) | 50.5 (1.2) | 51.0 (5.0) |
| PCPs | **88.7** (0.5) | **52.8** (0.1) | **63.5** (0.7) |

Table 1: **Patient cardiac prototypes outperform state-of-the-art core-set construction methods.** For Chapman and PTB-XL, we train an SVM, whereas for CPSC we train a random forest (due to multi-label classification). The core-set size equals the total number of PCPs. Mean and standard deviation are shown across five seeds. Since the raw instances of PTB-XL are 12-lead ECG signals, they could not be used with an SVM.

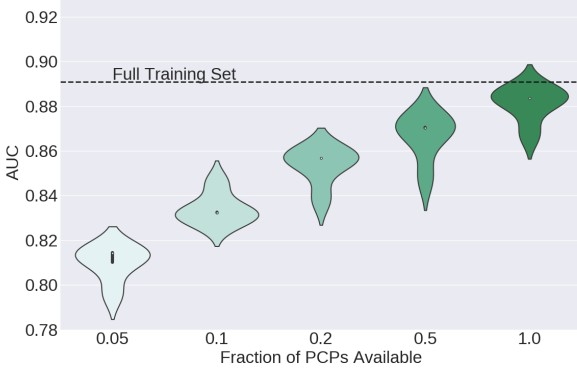

Figure 5: **Patient cardiac prototypes are effective dataset distillers.** Results are averaged across five random seeds. The horizontal dashed line depicts the performance of an SVM trained on all *instances* ($N = 76644$) in the training set. Despite a 12-fold reduction in the number of training instances ($F = 1$, $\Omega_{train} = 6,387$), the SVM achieved similar performance (AUC $\approx 0.89$) to one trained on all instances.

that which is redundant for solving the task at hand. We also observed that more extreme distillation does not significantly hinder the performance of PCPs. For example, an SVM model trained with only 5% of the available PCPs ($\Omega_{train} = 319$) achieved AUC $\approx 0.82$. Expressed differently, this corresponds to a 7% drop in performance despite a *240-fold* reduction (relative to training on all instances) in the number of training instances provided to the model. Such a finding reaffirms the potential of PCPs as dataset distillers.

### 4.7 Patient cardiac prototypes reliably retrieve similar patient data

**Quantitative evaluation**   We use PCPs as queries to search for and retrieve patient data, and present the precision and negative predictive value of the retrieved patient data as a function of the mathematical similarity between the PCPs and patient data, quantified via either Euclidean distance (Fig. 6 left) or cosine similarity (Fig. 6 right).

We found that patient cardiac prototypes reliably retrieved patients with a similar cardiac arrhythmia label. This is evident by the high precision achieved by PCPs at low Euclidean distance (high cosine similarity) values (Fig. 6). For example, $> 90\%$ of the pairs of patients that were deemed very similar to one another by our framework (i.e., $d_E < 6.2$) exhibited the same exact cardiac arrhythmia label. We also found that the precision decays as patients are deemed less similar to one another. For example, as $d_E \to 8.5$, the Precision $\to 0.3$. This finding, which is promising and expected from a reasonable similarity metric, is also exhibited by the precision curve in Fig. 6 (right). Note that, in this case, the precision curve is flipped along the y-axis since the most similar patients are those with the largest cosine similarity.

Based on these findings, we can identify a threshold distance between pairs of patients beyond which we are guaranteed a particular precision. This is useful for end-users looking for an empirical upper bound on the error. For example, in Fig. 6 (left), given a user-defined acceptable level of precision (e.g., 0.90), we identify the threshold distance (e.g., $d_E \approx 6.2$) below which patients have a high probability of being similar. This naturally lends itself to a *region of similarity*, where patients identified as being similar are highly likely of actually being so.

**Qualitative evaluation**   To provide a qualitative understanding of the retrieval capabilities of our framework, we produced a matrix reflecting the distance (or similarity) between each pair of patients before retrieving the most similar pair of patients. We present their associated data in Fig. 7.

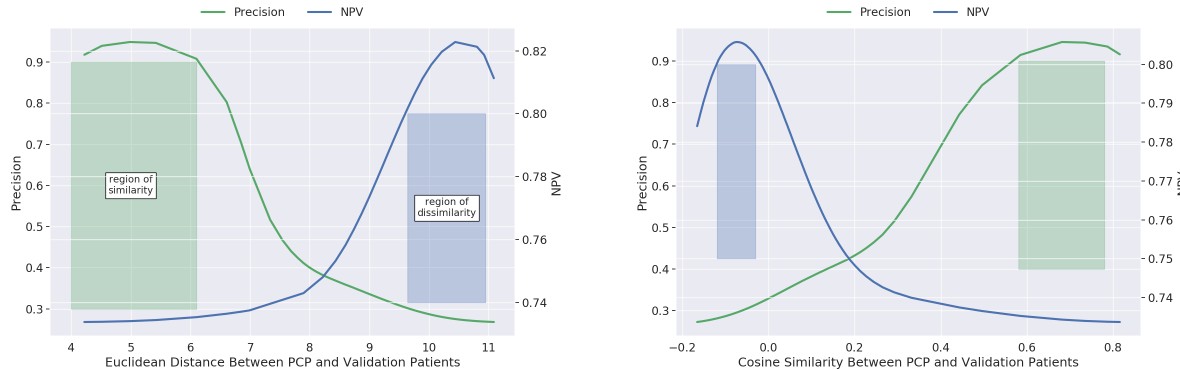

Figure 6: **Patient cardiac prototypes reliably retrieve similar patient data.** The similarity (or dissimilarity) metric used is the (left) Euclidean distance or (right) cosine similarity between PCPs and patient data in the validation set. We define the relevance of the retrieved patient data based on whether their corresponding cardiac arrhythmia class matches (or does not match) that of the query PCP. We show that our framework is agnostic to the type of similarity metric (e.g., Euclidean distance, cosine similarity) used for retrieval.

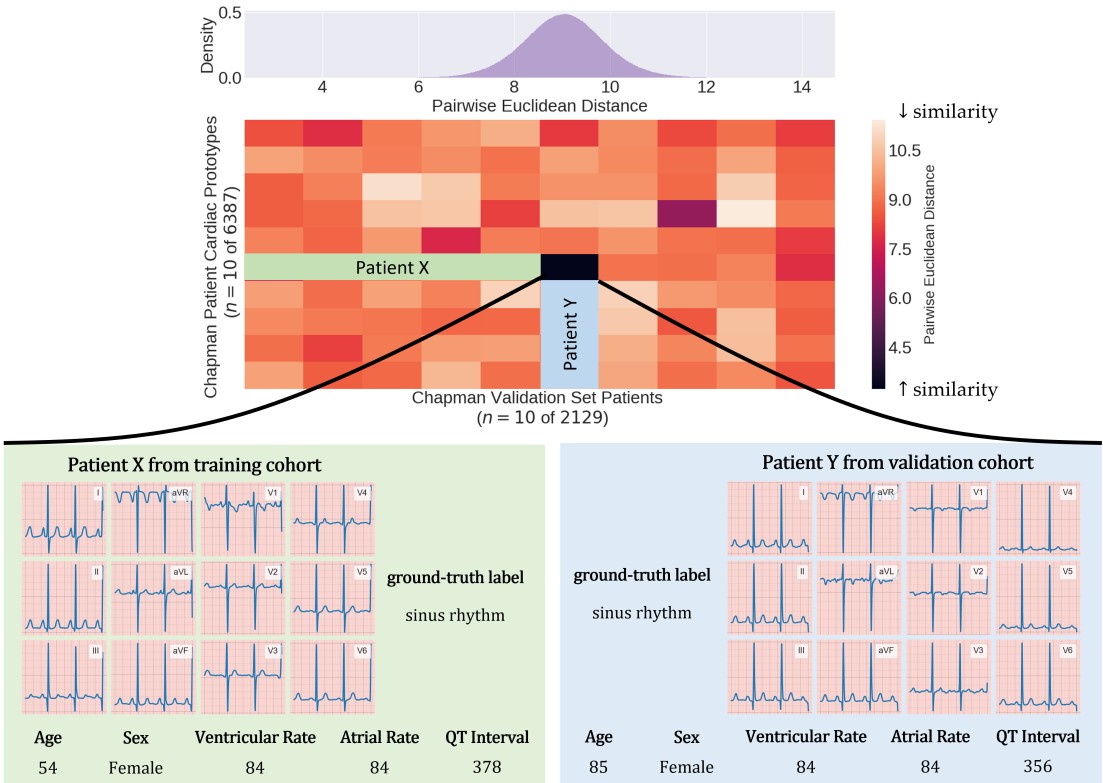

Figure 7: **Patient cardiac prototypes retrieve patients who have similar attributes.** We show a distribution of pairwise Euclidean distances between PCPs and representations in the validation set. We also present a subset of these pairwise distances in a matrix reflecting patient-patient distance values. We identify the most similar patient pair (↓ Euclidean distance) and retrieve their corresponding 12-lead electrocardiogram recordings, and where available, additional patient information.

We found that PCPs were able to sufficiently distinguish between unseen patient data and thus act as reasonable patient data similarity tools. This is evident by the large range of distance values for any chosen PCP (any matrix row, Fig. 7). In other words, PCPs were closer to some representations than to others, implying that a chosen PCP was not trivially equidistant to all other representations. However, distinguishing between patient data is not sufficient for a patient data similarity tool. We found that PCPs can also correctly retrieve relevant patient data. We found that the two patients identified as being most similar to one another, using our method, do indeed share many similarities (Fig. 7, bottom). For example, their respective 12-lead ECG data are both associated with the cardiac arrhythmia label of sinus rhythm. Furthermore, similarities are observed when comparing the cardiac-specific statistics such as ventricular rate (84 in both cases) and atrial rate (84 in both cases). We hypothesize that this behaviour arises due to the ability of PCPs to efficiently summarize the cardiac state of a patient, a finding which reaffirms the potential of PCPs as tools for patient data retrieval. We observed similar findings on the remaining datasets (see Appendix F).

## 5   Discussion

Our study was inspired by how primary care physicians adapt population-based protocols to the unique patient under their care. To that end, we proposed a framework which learns embeddings, titled patient cardiac prototypes, that capture the information unique to a patient's cardiac data. We demonstrated that PCPs, when incorporated into the diagnosis pipeline, can allow for the probing of deep learning-based

medical diagnoses. We also showed that PCPs are effective dataset distillers, where they can be used to train models that match the performance of those trained on a full dataset, which is orders of magnitude larger. We also demonstrated that PCPs can reliably retrieve similar patients across clinical databases.

Our framework has the potential to improve the transparency of clinical deep learning systems, providing further insight into why a deep learning-based medical diagnosis was made for an individual patient. This transparency is likely to instil clinical stakeholders with trust, and engage them further on the path to the deployment of such systems within clinical ecosystems. We believe that these prototypes naturally extend to other disciplines of medicine in which data from a particular cohort can be succinctly summarized in embedding form. Furthermore, as an effective approach to dataset distillation, our framework has the potential to reduce the resources and computational requirements for training clinical deep learning systems. Instead of training models on full datasets, in their raw format, researchers can now share and learn from prototypes directly (which can be orders of magnitude smaller in size than full datasets). Additionally, as clinical data are generated at an ever-growing pace, prototypes allow researchers to search for and quickly retrieve patient data considered relevant for their use-case (e.g., clinical trial enrolment). For example, recent work demonstrated that prototypes can be learned with additional patient attributes (e.g., sex and age) as a means of retrieving and annotating previously-unlabelled cardiac signals (Kiyasseh et al., 2021a).

Our study does suffer from several limitations. Despite the multi-purpose use of patient cardiac prototypes, they remain relatively myopic; they do not account for diverse spatial and long-range temporal changes in patient data. When learning PCPs, we implicitly assumed that all representations of cardiac signals belonging to the same patient should be attracted to a single PCP. This is a valid assumption if such representations do indeed reflect a similar underlying physiological state. However, this assumption may not hold if patient data span multiple years or pertain to distinct modalities (e.g., imaging, electronic health records, etc.). In such a setting, patient cardiac prototypes would be a spatial and temporal average of a patient's clinical state. Although this summary embedding could be of use in certain scenarios, it might conceal subtle but useful changes in the patient's physiological state.

Although we demonstrated that patient cardiac prototypes have multiple applications, we did observe that a system learned without a contrastive loss (i.e., without prototypes) performed marginally better (AUC = 0.90) than one learned with such a loss (AUC = 0.88). It is unknown whether this difference leads to meaningful and noticeable changes within the clinic. We leave it to future work to explore other variants of contrastive learning (Bardes et al., 2022). More broadly, however, it remains an open question whether there exists a trade-off between system transparency and performance, and in such an event, whether clinical stakeholders are willing to sacrifice some performance in exchange for the improved transparency and widened application areas of deep learning systems.

Moving forward, and as data become available, we aim to learn prototypes over longer time-spans (e.g., years) and from multiple data modalities. Such temporal multi-modal prototypes would allow researchers to (a) monitor changes in the clinical state of a patient over time and (b) obtain a more holistic summary of that state. For example, a single multi-modal prototype might capture the information from coronary angiogram data, an electrocardiogram, and a clinical report all collected on the same day (e.g., at an annual checkup). A distinct prototype can then be learned for every subsequent annual checkup, ultimately providing a succinct temporal trajectory of a patient's clinical state. Such prototypes may further propel the diagnostic performance of clinical deep learning systems and inform downstream temporal analyses (e.g., survival analyses).

We also aim to exploit the quantification of the similarity of patient data to advance graph neural networks. Such networks typically require the design of an adjacency matrix, one that quantifies the presence and weight of edges (relationships) between nodes (patients) (Li et al., 2022). However, designing this matrix is non-trivial, particularly when dealing with physiological data, and can become a computational burden if dense. By interpreting these nodes as patients, our PCP-derived similarity values can be used to *initialize* the weights of edges between nodes and potentially inform the sparsity of node connections. As such, graph neural networks can be thought of as being trained with a PCP-derived prior, one that could accelerate, and inject valuable domain knowledge into, the learning process.

## 6 Conclusion

Clinical deep learning systems are becoming increasingly adept at automating medical diagnoses. However, the vast majority of these systems continue to make predictions that are population-based and opaque. Here, we introduced a framework anchored around the concept of patient cardiac prototypes, learned embeddings which uniquely capture the information of an individual patient's cardiac data. We hope the community joins us in building upon these prototypes in other application areas to improve the trustworthiness of clinical deep learning systems and increase their likelihood of adoption by clinical stakeholders.

**Acknowledgements**

We thank the anonymous reviewers for their insightful feedback. We also thank Talal Maddah for lending us his voice. David Clifton was supported by the EPSRC under Grants EP/P009824/1 and EP/N020774/1, and by the National Institute for Health Research (NIHR) Oxford Biomedical Research Centre (BRC). The views expressed are those of the authors and not necessarily those of the NHS, the NIHR or the Department of Health. Tingting Zhu was supported by the Engineering for Development Research Fellowship provided by the Royal Academy of Engineering.

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

## Appendix

## A Further motivation behind spatial and temporal invariances

We acknowledge that, in some cases, intra-patient invariance *can* be undesirable. An example is when an individual patient's cardiac signals are collected over a large time-span (e.g., on the order of years), during which their physiological state might have changed, and are thus likely to reflect distinct information. By collapsing this distinct information into a single embedding, as is done with PCPs, we would lose diagnostically relevant insight. From a practical perspective, and in our experiments, we avoid this problematic scenario by exclusively considering data points that are on the order of seconds (e.g., 10 seconds) and which are collected during the same hospital visit (see Fig. 1 left). To achieve this, we used the patient ID meta information, and, where available, the date of the ECG recording. An intra-patient invariance is therefore likely to hold in light of the (a) short time-span over which the data recording took place and (b) low likelihood of an electrocardiogram signal exhibiting major morphological changes during this time.

## B Datasets

### B.1 Data pre-processing

For all of the datasets, frames consisted of 2500 samples and consecutive frames had no overlap with one another. Data splits were always performed at the patient-level.

**CPSC** (Alday et al., 2020). Each ECG recording varied in duration from 6 seconds to 60 seconds with a sampling rate of 500Hz. Each ECG frame in our setup consisted of 2500 samples (5 seconds). We assign multiple labels to each ECG recording as provided by the original authors. These labels are: AF, I-AVB, LBBB, Normal, PAC, PVC, RBBB, STD, and STE. The ECG frames were normalized in amplitude between the values of 0 and 1.

**Chapman** (Zheng et al., 2020). Each ECG recording was originally 10 seconds with a sampling rate of 500Hz. We downsample the recording to 250Hz and therefore each ECG frame in our setup consisted of 2500 samples. We follow the labelling setup suggested by Zheng et al. (2020) which resulted in four classes: Atrial Fibrillation, GSVT, Sudden Bradychardia, Sinus Rhythm. The ECG frames were normalized in amplitude between the values of 0 and 1.

**PTB-XL** Wagner et al. (2020). Each ECG recording was originally 10 seconds with a sampling rate of 500Hz. We extract 5-second non-overlapping segments of each recording generating frames of length 2500 samples. We follow the diagnostic class labelling setup suggested by Wagner et al. (2020) which resulted in five classes: Conduction Disturbance (CD), Hypertrophy (HYP), Myocardial Infarction (MI), Normal (NORM), and Ischemic ST-T Changes (STTC). We alter the original setup in two main ways. Firstly, we only consider ECG segments with one label assigned to them. Secondly, we convert the task into a binary classification problem of NORM vs. (CD, HYP, MI, STTC) from above. The ECG frames were normalized in amplitude between the values of 0 and 1.

### B.2 Data samples

In this section, we outline the number of instances used during training, validation, and testing for the CPSC, Chapman, and PTB-XL datasets.

| Dataset | Train | Validation | Test |
|---------|-------|------------|------|
| CPSC | 157,188 (4,402) | 37,296 (1,100) | 47,460 (1,375) |
| Chapman | 76,614 (6,387) | 25,524 (2,129) | 25,558 (2,130) |
| PTB-XL | 286,632 (10,807) | 36,816 (1,411) | 37,008 (1,383) |

Table 2: Number of instances (number of patients) used during training. These represent sample sizes for all 12 leads.

## C  Further implementation details

### C.1  Network architecture

In this section, we outline the architecture of the neural network used for all experiments.

| Layer Number | Layer Components | Kernel Dimension |
|:---:|:---:|:---:|
| 1 | Conv 1D
BatchNorm
ReLU
MaxPool(2)
Dropout(0.1) | 7 x 1 x 4 ($K$ x $C_{\text{in}}$ x $C_{\text{out}}$) |
| 2 | Conv 1D
BatchNorm
ReLU
MaxPool(2)
Dropout(0.1) | 7 x 4 x 16 |
| 3 | Conv 1D
BatchNorm
ReLU
MaxPool(2)
Dropout(0.1) | 7 x 16 x 32 |
| 4 | Linear
ReLU | 320 x $E$ |
| 5 | Linear | $E$ x C (classes) |

Table 3: Network architecture used for all experiments. $K$, $C_{\text{in}}$, and $C_{\text{out}}$ represent the kernel size, number of input channels, and number of output channels, respectively. A stride of 3 was used for all convolutional layers. $E$ represents the dimension of the final representation.

| Dataset | Batchsize | Learning Rate |
|:---:|:---:|:---:|
| CPSC | 256 | $10^{-4}$ |
| Chapman | 256 | $10^{-4}$ |
| PTB-XL | 256 | $10^{-3}$ |

Table 4: Batchsize and learning rates used for training with different datasets. The Adam optimizer was used for all experiments.

### C.2  Baseline methods

In constructing a core-set, the baseline methods (Lucic (Lucic et al., 2016), Lightweight (Bachem et al., 2018), and Archetypal (Mair & Brefeld, 2019)) typically followed a similar strategy. These methods generated a categorical proposal distribution over all instances in the dataset before sampling $k$ instances and assigning them weights. For a fair comparison to patient cardiac prototyps (PCPs), we chose $k = P$ where $P$ is the number of PCPs.

# D    Description of the ablation studies

To gain a better understanding of the marginal benefit of each of our framework's components on its overall performance, we conducted an extensive set of ablation studies, as outlined next.

## D.1    Experimenting with variants of the retrieval mechanism

During inference on an unseen data point, our framework retrieves patient cardiac prototypes learned on exclusively on the *training* dataset (see Retrieving PCP during inference). We experimented with variants of this retrieval mechanism as part of an ablation study, which we outline next.

**Retrieval variant 1 (mean)**    The first variant, which we refer to as *mean*, involved taking the average of all PCPs, $\bar{\boldsymbol{p}} = \Omega_{train}^{-1} \cdot \sum_{j=1}^{\Omega_{train}} \boldsymbol{p}_j$, regardless of the instance in the held-out set. Since PCPs and the hypernetwork are deterministic during the inference stage, this approach implies that the generated linear parameters are effectively reduced to a constant.

**Retrieval variant 2 (similarity-weighted mean)**    The second variant, which we refer to as *similarity-weighted mean*, involved calculating a linear combination of the PCPs, weighted according to their similarity to the representation, $\boldsymbol{h}_i$, of an instance. Formally, $\bar{\boldsymbol{p}} = \sum_{j=1}^{\Omega_{train}} s(\boldsymbol{p}_j, \boldsymbol{h}_i) \cdot \boldsymbol{p}_j$. In effect, this approach exploited information unique to a patient's ECG data to down-weight, or up-weight, the contribution of each PCP.

**Retrieval variant 3 (nearest)**    The third variant, which we refer to as *nearest*, involved the vanilla approach of retrieving the *single* PCP closest to the representation, $\boldsymbol{h}_i$. In effect, this approach retrieved the PCP of the patient in the training set that was deemed most similar to the representation of an instance associated with a different patient in the held-out set.

**Retrieval variant 4 (nearest 10)**    We hypothesized that by restricting the framework to only select a single PCP, we were preventing the representation from exploiting potentially useful information contained in additional PCPs. For example, these additional PCPs could reflect patients with attributes (e.g., sex, age, and treatment outcome) that are shared with, and thus potentially useful for, the patient in the held-out set for whom the prediction is being made. Therefore, our fourth variant of the retrieval mechanism, which we refer to as *nearest 10*, involved taking the average of the ten PCPs closest to the representation, $\boldsymbol{h}_i$.

## D.2    Experimenting with variants of the patient cardiac prototypes

To gain some better intuition as to whether PCPs were indeed capturing information unique to patient ECG data, we learned embeddings that differed slightly from our patient cardiac prototypes.

**Optimizing the supervised loss alone**    In this ablation study, we trained our deep learning framework *without* a contrastive loss term (equation 1) and, therefore, did not learn the corresponding PCPs. In this scenario, we treated the average representation from each patient (instead of the PCP) as the descriptor of an individual patient's ECG data. In the Results, we refer to these embeddings as Mean Representations.

**Optimizing the contrastive loss alone**    In this ablation study, we exclusively optimized the InfoNCE loss. Here, supervision manifested solely in the form of patient IDs. Importantly, cardiac arrhythmia disease labels are *not* seen by the network during training. After learning prototypes in this setting, we exploited them to perform cardiac arrhythmia classification. In contrast to previous experimental settings, this setting does not include a linear classification head that would have output a probability distribution over the possible classes. Therefore, during inference, we decided to implement the k-nearest neighbours (knn) algorithm as a way to mimic the previously-introduced retrieval mechanism. Specifically, for each representation in the held-out set, we searched for its $K$ nearest prototypes (using cosine similarity) and considered the most frequent cardiac arrhythmia ground-truth label of these prototypes as the diagnosis. To clarify, $K = 1$ and

$K = 10$ can be thought of as being analogous to the nearest and nearest 10 retrieval mechanisms introduced earlier.

## E    Further description of the patient data retrieval process

Patient cardiac prototypes have the potential to retrieve similar patient data within clinical databases. To substantiate this claim, we followed these steps.

**Step 1 - calculate distance between patients**    When we searched for cardiac signals *within a dataset*, we first calculated the Euclidean distance, $d(\boldsymbol{p}_j, \boldsymbol{h}_i)$ between the $j$-th PCP, $\boldsymbol{p}_j$, and the representation, $\boldsymbol{h}_i$, of the $i$-th instance unseen during training (e.g., those in the validation set). Our motivation for choosing the Euclidean distance, as opposed to some other similarity metric, such as cosine similarity, will be discussed later. At this point, we had access to distances between a particular patient (in the form of a PCP) and representations. However, to obtain distances between patients and other patients, we averaged these distance values across representations, $\boldsymbol{h}_i$, of instances associated with the same patient. This process was then repeated for all PCPs. In contrast, when we searched for cardiac signals *across distinct datasets*, we simply calculated the Euclidean distance between the PCPs of these respective datasets. This immediately provided us with patient-patient distance values.

**Step 2 - define relevance of retrieved patients**    Evaluating the relevance of the retrieved patient data is non-trivial. This is because the similarity (and dissimilarity) of patient data from a clinical perspective is nebulous. For example, patient data can be deemed similar based on attributes such as sex and age, medical history (e.g., cancer survivor), and drug treatment pathways. Unfortunately, publicly-available datasets of cardiac signals do not contain such exhaustive information. However, these datasets do entail cardiac arrhythmia disease labels. Therefore, each pair of patients was assigned a ground-truth relevance score ($s = 1$ relevant, $s = 0$ irrelevant) according to whether they shared the same cardiac arrhythmia label. We then identified pairs of patients as being relevant if their Euclidean distance, $d < d_E$, was less than some threshold distance, $d_E$. If, however, we were interested in retrieving dissimilar patients, then we would simply need to redefine the ground-truth relevance score and identify pairs of patients with $d > d_E$ as being relevant (notice the swap in the sign).

For the precision metric, we were looking to quantify how many of the retrieved patients (those identified as relevant) were actually relevant. In contrast, for the NPV, we were looking to quantify how many of the retrieved patients (those identified as irrelevant) were actually irrelevant.

# F  Additional results

## F.1  Effect of supervision on performance

As part of our ablation studies, we examined the effect of removing the supervision outlined in equation 2 on the overall performance of our framework. We refer to this setup at NCE Loss Only since we only end up optimizing the InfoNCE loss. The prototypes learned in such a manner are similar in spirit to those learned by Li et al. (2021) in an unsupervised manner. We use these prototypes as part of a K-nearest neighbours model to classify cardiac arrhythmias. As is apparent in Fig. 8, we found that these prototypes, which are learned in an unsupervised manner, perform worse (AUC < 0.60) than those learned with supervision (AUC > 0.75). This is expected as the supervision in equation 2 is directly related to the cardiac arrhythmia class. We suspect that other purely unsupervised contrastive learning methods would perform equally poorly.

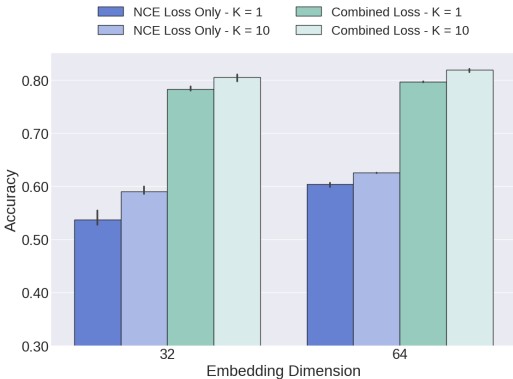

Figure 8: Comparison of the performance of the $K$-nearest neighbours model with embeddings learned by exclusively optimizing the InfoNCE loss (NCE Loss Only) against that of our framework (Combined Loss). We show that such embeddings are less predictive of cardiac arrhythmia classes than the prototypes learned via our proposed framework.

### F.2  Effect of InfoNCE Loss on Dataset Distillation

In this section, we explore the effect of exclusively optimizing the InfoNCE loss on dataset distillation. Specifically, we train our network to optimize the InfoNCE loss (without the supervised cross-entropy loss) and learn prototypes in an end-to-end manner. To evaluate the dataset distillation capabilities of these learned prototypes, we exploit them to train a support vector machine (SVM) to classify cardiac arrhythmia classes. The intuition is that if these prototypes happen to be effective dataset distillers, then we would models trained exclusively on them to perform equally well (on a held-out dataset) as models trained on the entire dataset.

In Fig. 9, we present the accuracy of the SVM models trained on these prototypes as a function of the fraction of prototypes available for training. We show that prototypes learned in such a manner are *ineffective* dataset distillers; the performance of SVM models trained exclusively on such prototypes is worse than that achieved with a model trained on all representations. For example, at $F = 1$, these two models achieve Accuracy $\approx 0.40$ and 0.60, respectively. We hypothesize that this poor performance (across all fractions) is due to the lack of class-discriminative behaviour exhibited by the learned prototypes. In other words, there exists a weak mapping from prototypes to cardiac arrhythmia classes. Such a finding suggests that, from the perspective of dataset distillation, our proposed combined loss (see Methods section of main manuscript) is preferable to the InfoNCE loss.

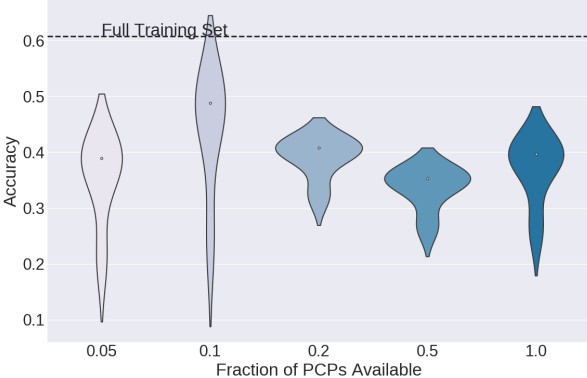

Figure 9: **AUC of SVM model trained on a fraction, $F$, of the prototypes and evaluated on a held-out set of data.** Results are averaged across five random seeds. The horizontal dashed line depicts the performance of a model trained on all of the training representations. The prototypes are initially learned via a framework which optimizes only the InfoNCE loss. We show that prototypes learned in such a manner are ineffective dataset distillers; the performance of SVM models trained exclusively on such prototypes is worse than that achieved with a model trained on all representations.

### F.3 Retrieving similar patients across datasets

In Fig. 10, we illustrate the pair of patients (across distinct datasets) identified as being most similar to one another based on our PCP framework. We also present the 12-lead electrocardiogram data for this pair of patients in order to help validate our patient retrieval mechanism. We find that PCPs can reliably retrieve patients *across* distinct datasets that are similar to one another. This is evident by the similar morphology exhibited by the pair of 12-lead ECG data. For example, both patients exhibit normal electrical heart activity, which is also known as normal sinus rhythm.

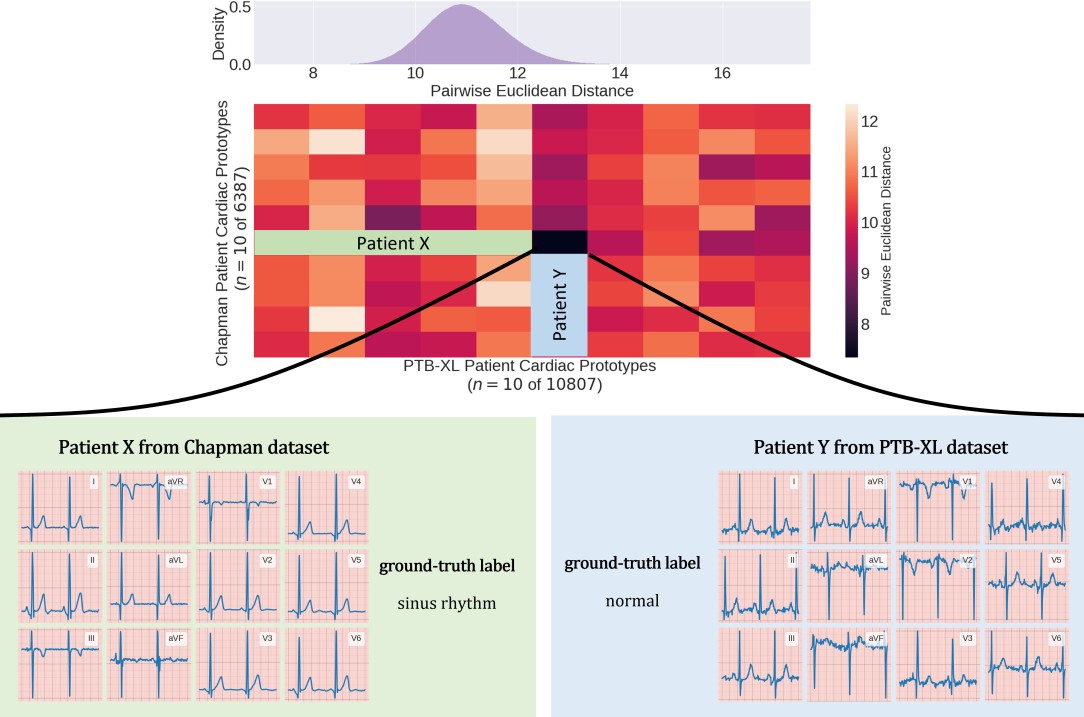

Figure 10: **Exploiting PCPs to discover similar patients across the Chapman and PTB-XL datasets.** We show a distribution of pairwise Euclidean distances between PCPs and representations in the validation set. We also present a subset of these pairwise distances in a matrix reflecting patient-patient distance values. We identify the most similar patient pair ($\downarrow$ Euclidean distance) and retrieve their corresponding 12-lead electrocardiogram recordings. We show that PCPs can reliably identify similar patients. This is evident by the high degree of similarity exhibited by the pair of patients. For example, both patients exhibit a cardiac arrhythmia label of sinus rhythm.

### F.4 Retrieving dissimilar patients across datasets

In addition to showing that PCPs can be used to retrieve similar patients, we claim that they can also be exploited for the retrieval of *dissimilar* patients. In this section, we provide qualitative evidence in support of this claim. In Fig. 11 (top), we present a pair of patients within the same dataset identified as being most dissimilar from one another. In Fig. 11 (bottom), we present a pair of patients *across* distinct datasets identified as being most dissimilar from one another. In both cases, we find that PCPs can reliably retrieve dissimilar patients. This is evident by the observation that the morphology of the pair of the 12-lead ECG data differs. For example, in Fig. 11 (top), the two dissimilar patients exhibit a normal rhythm (sinus rhythm) and a potentially fatal one (atrial fibrillation).

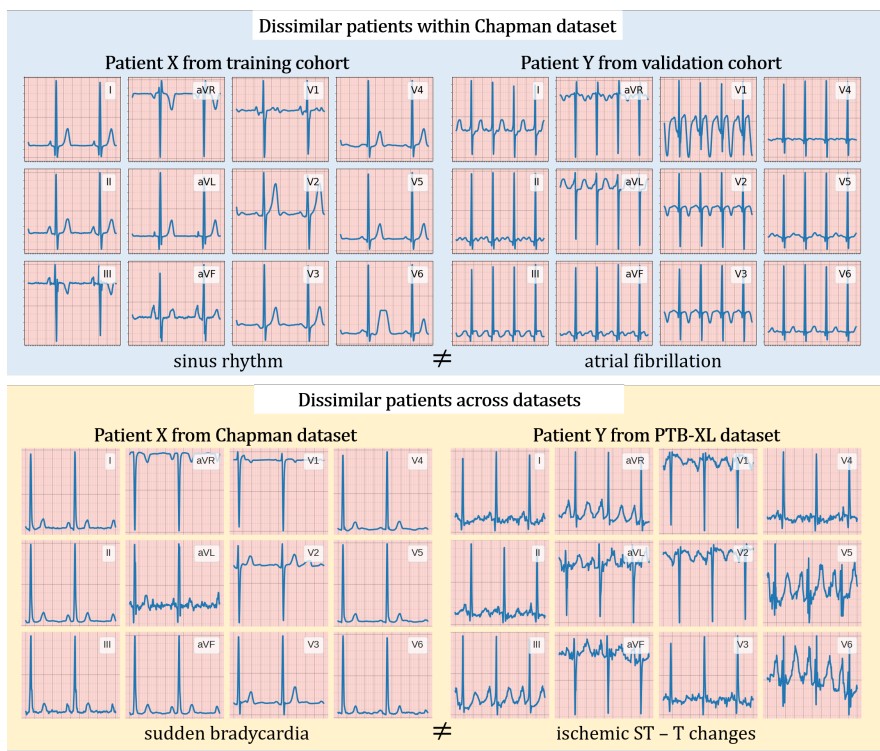

Figure 11: **12-Lead ECG segments corresponding to a pair of patients identified as being dissimilar from one another based on the PCPs.** Dissimilarity is defined as a high Euclidean distance between patient cardiac prototypes (PCPs) and representations of instances in the validation set. We show that PCPs can reliably retrieve dissimilar patients. This is evident by the observation that the ECG segments between patients exhibit different morphology and correspond to the different cardiac arrhythmia labels (sudden bradycardia vs. ischemic ST-T changes).

