# OpenReview forum: "PCPs: Patient Cardiac Prototypes to Probe AI-based Medical Diagnoses, Distill Datasets, and Retrieve Patients"
_TMLR — Accepted by TMLR_

### Review · Reviewer_9duA · 2022-11-14

**Summary Of Contributions:**

Firstly, I would like to highlight that the paper gets better as it progresses through to the experimental section. Before that the paper is a bit shallow in that it should be more concise. Secondly the paper provides a framework to distill information from raw data into something they term patient cardiac prototypes (PCP), given they are using ECGs. They argue that their framework, that is based upon contrastive learning, can generate embeddings (i.e. PCPs) that can replace the raw data (i.e. ECGs) which are more volume/size demanding. The overarching aim is to develop a more interpretable way of doing inference, hence being able to refer back to the original PCP that contributed to the outcome. In addition, the process involves the development of a hyper network that gets an input representation from there feature extractor which then provides an input to the linear classification layer. Lastly the authors provide a set of experiments to show how PCPs can be used to detect cardiac arrythmia, as well as a set of ablation studies to show how the proposed framework contributes to the overall performance and interpretability in the diagnosis.

**Audience:**

Yes

**Broader Impact Concerns:**

Not an ethical concern per we, but it could be that intepretability arguments with respect to medical diagnosis might be better tackle with the involvement of clinicians and/or additional experiments.

**Claims And Evidence:**

Yes

**Requested Changes:**

— To my understanding, at the inference stage, the unseen input is associated with a specific PCP but at the same time it lends characteristics from the wider representation space. Is that accurate? If so, I am curious as to how well this generalises in cases beyond the PCPs proposed here, i.e. whether that is observed in other cases as well. Are figures 6, 7 supposed to illustrate this fact?

— The literature review is rather brief, e.g. what studies exist in this area, and also what other self-supervised learning approaches could be relevant, i.e. why contrastive learning here?

— Discussion could be better organised to provide a section for conclusion.

**Strengths And Weaknesses:**

Strengths:

— the experimental setup is appropriate
— the ablation studies are informative, including visualisations
— overall the presentation is fine, but some typos exist, e.g. in contribution number 4.

Weaknesses

— although the presentation is fine, the writing up could be improved in that one could flesh out what the technical contributions of this paper are. InfoNCE loss is not cited when it is first used, so I think the 2018 paper that proposed it (contrastive Predictive Coding; van den Oord, et al. 2018).
— The intermediate embeddings are not different to any other embeddings one can develop with raw data, but it is not clear to me how the contrastive component helps to develop the PCPs in a way that one can trace the outcome back to the single PCP based on some Euclidean metric - more explanation would be useful
— I am not a clinician therefore my opinion should be taken with a pinch of salt, but the second sentence in the abstract might not be accurate. In certain countries, GPs or other physicians do follow population-based protocols to make decisions for individuals, albeit sometimes adapting to individual circumstances. However it is not always individualised, hence precision medicine is an area that people are looking towards for the future.

---

> ### Author Response · Authors · 2022-11-25
> **Response to Reviewer 9duA (Part 1)**
>
> ### **Comment 1**
> Although the presentation is fine, the writing up could be improved in that one could flesh out what the technical contributions of this paper are.
>
> #### **Response to Comment 1**
> To clearly outline the contributions of our paper, we have now modified the contributions paragraph in the Introductions section (page 2, paragraph 1).
>
> ### **Comment 2**
> InfoNCE loss is not cited when it is first used, so I think the 2018 paper that proposed it (contrastive Predictive Coding; van den Oord, et al. 2018)
>
> #### **Response to Comment 2**
> We have now cited this paper upon our first mentioning of the InfoNCE loss (page 2, last paragraph)
>
> ### **Comment 3**
> The intermediate embeddings are not different to any other embeddings one can develop with raw data, but it is not clear to me how the contrastive component helps to develop the PCPs in a way that one can trace the outcome back to the single PCP based on some Euclidean metric - more explanation would be useful
>
> #### **Response to Comment 3**
> To better understand how a medical diagnosis can be traced back to a single PCP, we must distinguish between the training and inference stages of our framework. During the inference stage, and while leveraging the nearest retrieval mechanism, we select the single PCP which is closest (lowest Euclidean distance) to the representation of the unseen input (see Fig. 1, right). As such, the final outcome can be loosely attributed to this retrieved PCP.
>
> During the training stage, we simply needed a way to learn patient-specific embeddings (i.e., PCPs). In other words, we wanted to learn embeddings that reflected the cardiac state of a patient. One way to do so is via supervised contrastive learning, which is the approach we adopted. Therefore, how the embeddings are learned (e.g., supervised contrastive learning) is independent of the inference mechanism (e.g., retrieval of embeddings and use of hypernetwork). This implies that other researchers may decide to learn patient-specific embeddings differently from us but continue to use our same proposed inference stage.
>
> To clarify this matter, we have now created an analogy between quantifying the similarity of patients and (a) the attention mechanism of Transformers and (b) edges between nodes in a graph neural network (page 3, last paragraph), where the latter is reflected in the modified Figure 1 (page 4). Specifically, we state “This similarity between patients is akin to the cross-attention mechanism used in Transformer architectures or to edges connecting patients in a graph.”
>
> ### **Comment 4**
> I am not a clinician therefore my opinion should be taken with a pinch of salt, but the second sentence in the abstract might not be accurate. In certain countries, GPs or other physicians do follow population-based protocols to make decisions for individuals, albeit sometimes adapting to individual circumstances. However it is not always individualised, hence precision medicine is an area that people are looking towards for the future.
>
> #### **Response to Comment 4**
> We appreciate the precision of the reviewer’s statement relative to ours and have thus modified the second sentence in the abstract to reflect that sentiment. It now states “This is in contrast to how primary care physicians make decisions, often adapting population-based protocols to the unique patient under consideration.”

---

> > ### Author Response · Authors · 2022-11-25
> > **Response to Reviewer 9duA (Part 2)**
> >
> > ### **Comment 5**
> > To my understanding, at the inference stage, the unseen input is associated with a specific PCP but at the same time it lends characteristics from the wider representation space. Is that accurate? If so, I am curious as to how well this generalises in cases beyond the PCPs proposed here, i.e. whether that is observed in other cases as well. Are figures 6, 7 supposed to illustrate this fact?
> >
> > #### **Response to Comment 5**
> > To clarify, during the inference stage, an “unseen” ECG input refers to a sample from a patient whose data were not presented to the deep learning system during the training stage. In light of this split in the patient data across the training and inference stages, we are in essence evaluating the degree to which the deep learning system generalises to new patients.
> >
> > We provided evidence in favour of such generalisation in the context of medical diagnosis, dataset distillation, and patient retrieval. For medical diagnosis, the results in Fig. 4 (top, Nearest 10) depict the deep learning system’s ability to generalise to patients in the validation set (AUC > 0.85) and whom did not appear in the training set whatsoever. For dataset distillation, the results in Fig. 6 demonstrate that after training exclusively on PCPs, the deep learning system can also generalise to patients in the validation set (AUC > 0.80). For patient retrieval, the results in Fig. 7 show that PCPs can be used to retrieve similar (or dissimilar) patients who are exclusively in the validation set. For further qualitative evidence, we direct the reviewer to Fig. 10 (Appendix F.3) where we demonstrate that PCPs, in the context of patient retrieval, can also generalise to patients in an entirely different dataset (as evident by the retrieval of patients with similar cardiac conditions, namely normal sinus rhythm).
> >
> > ### **Comment 6**
> > The literature review is rather brief, e.g. what studies exist in this area, and also what other self-supervised learning approaches could be relevant, i.e. why contrastive learning here?
> >
> > #### **Response to Comment 6**
> > We have now expanded the paragraph outlining previous work (page 1, paragraph 3), notably to include references to similar approaches with graph neural networks. As for motivating the contrastive learning approach, in Section 2.1 (page 2), we provide a description of the main study which inspired our current approach and why we believe learning invariances, through contrastive learning, can confer benefits.
> >
> > ### **Comment 7**
> > Discussion could be better organised to provide a section for conclusion.
> >
> > #### **Response to Comment 7**
> > We have now modified the Discussion section to outline the following in order: our main findings (paragraph 1), the implications of our study (paragraph 2), the limitations of our study (paragraph 3), remaining open questions (paragraph 4), and potential future work (paragraph 5). As suggested by the reviewer, we now include a Conclusion section with more high-level remarks.

---

### Review · Reviewer_FnHS · 2022-11-19

**Summary Of Contributions:**

This paper proposes a method to derive patient cardiac prototypes (PCP), which can be used for developing interpretable arrhythmia diagnosis system. In particular, the paper conducts the following experiments using three different ECG datasets to empirically validate the claimed contributions:
- Ablation study with diverse retrieval methods
- Ablation study with using only one loss term, instead of two
- Qualitative example of how PCP enables interpretable diagnosis
- Dataset distillation experiment
- Similar/dissimilar patient retrieval experiment


**Audience:**

Yes

**Claims And Evidence:**

No

**Requested Changes:**

The paper needs to rigorously address all weaknesses mentioned above. In my estimation, I see two possibilities:
- Carefully justify how retrieval-based diagnosis systems can help physicians in the real-world, when the physicians themselves don't even know the true label, and there is no way of knowing if the retrieved sample is indeed the most similar one to the given test sample, all the while the definition of "similar" is loosely defined. Then abandon prototypes because sample-wise retrieval can bring the same amount of interpretability as the prototype retrieval. Then demonstrate that sample-wise retrieval does not hurt diagnosis performance compared to none-retrieval-based models. This would naturally lead to abandoning all claims on dataset distillation.
- Withdraw any claims regarding interpretability. Stick to PCP, but only in the context of dataset distillation.

But there can always be a more creative solution, and as long as the weaknesses are resolved, I would be happy to re-evaluate this work.

**Strengths And Weaknesses:**

Strengths:
- The paper conducts an extensive set of experiments to evaluate the proposed method in multiple dimensions.
- If one is interested in learning a PCP, the suggested approach seems like a reasonable one to use.

Weakness:
- In Figure 4, one can see that PCP actually harms diagnosis accuracy. I suspect this would be even more evident if the paper used a more sensible baseline (e.g. a combination of 1D-Conv and Transformer or the winner of PhysionNet Challenge 2022), rather than simply using the proposed framework (Figure 1) minus the contrastive loss. If you are harming an AI's predictive accuracy to gain a little bit of interpretability, that AI has little chance to be used by real-world users, considering that a bad decision can risk human lives in healthcare.
- The paper claims that PCP can help physicians during the diagnostic process, because PCP is able to provide insight when the model makes both correct and incorrect decision. But I do not see how this is possible. How can PCP help physicians, when physicians themselves don't have GT labels during the test phase? What if the model makes an incorrect decision but the retrieved PCP looks similar enough to the test sample? Or conversely, what if the model makes a correct decision but the retrieved PCP does not look similar enough to the test sample?
- It's hard to see why we need prototypes to begin with. If we wanted to develop an interpretable diagnosis system based on retrieving the most similar training sample (although it is unclear whether this actually helps physicians, as mentioned above), we can perform sample-wise representation learning, and use individual samples for retrieval. I do not see why retrieving a patient-level prototype (instead of a single ECG sample) provides any additional value regarding interpretability. Prototypes might be good for dataset distillation, but that is a completely independent topic.
- The paper is overall very verbose. I see the same content repeatedly described multiple times throughout the paper. One example is how the paper repeatedly describe how they used short time-span ECG samples to ensure intra-patient invariance. The whole material of this paper does not seem to require 16 pages.

---

> ### Author Response · Authors · 2022-11-25
> **Response to Reviewer FnHS (Part 1)**
>
> ### **Comment 1**
> In Figure 4, one can see that PCP actually harms diagnosis accuracy. I suspect this would be even more evident if the paper used a more sensible baseline (e.g. a combination of 1D-Conv and Transformer or the winner of PhysionNet Challenge 2022), rather than simply using the proposed framework (Figure 1) minus the contrastive loss. If you are harming an AI's predictive accuracy to gain a little bit of interpretability, that AI has little chance to be used by real-world users, considering that a bad decision can risk human lives in healthcare.
>
> #### **Response to Comment 1**
> We appreciate the reviewer’s sentiment here. We would first like to clarify that our emphasis in this study was not to achieve state-of-the-art performance. Instead, our goal was to demonstrate the utility and applications of patient cardiac prototypes (e.g., for probing AI systems, dataset distillation, patient retrieval). Therefore, the utility of our study must be viewed holistically as it pertains to enabling all of these tasks (as opposed to just one).
>
> Having said that, we did observe that a system learned without a contrastive loss (and without prototypes) performed marginally better than one learned with a contrastive loss (AUC 0.90 vs. 0.88). It is unknown whether this difference leads to meaningful and noticeable changes within the clinic. More broadly, however, it remains an open question whether there exists a trade-off between system transparency and performance, and in such an event, whether clinical stakeholders are willing to sacrifice some performance in exchange for the improved transparency and widened application areas of deep learning systems.
>
> We have now included this in the Discussion section (page 11, paragraph 2).
>
> ### **Comment 2**
> The paper claims that PCP can help physicians during the diagnostic process, because PCP is able to provide insight when the model makes both correct and incorrect decision. But I do not see how this is possible. How can PCP help physicians, when physicians themselves don't have GT labels during the test phase? What if the model makes an incorrect decision but the retrieved PCP looks similar enough to the test sample? Or conversely, what if the model makes a correct decision but the retrieved PCP does not look similar enough to the test sample?
>
> #### **Response to Comment 2**
> We thank the reviewer for bringing this interesting thought experiment to our attention. Indeed, PCPs are unlikely to be used for decision support (i.e., assisting physicians during the diagnostic process). We have removed such claims from the manuscript.
>
> Our intention is to demonstrate that PCPs can allow researchers to probe deep learning-based predictions, as a way to potentially glean why a particular (mis)diagnosis was made. We have now included Section 4.4 (page 8) which reflects our intention more precisely and includes additional quantitative evidence (Figure 4, left) inspired by your suggested thought experiment. In short, we find that a high proportion of correct predictions are associated with retrieval of relevant PCPs. Conversely, a high proportion of incorrect predictions are associated with the retrieval of irrelevant PCPs (see Section 4.4 for definition of relevance).
>
> ### **Comment 3**
> It's hard to see why we need prototypes to begin with. If we wanted to develop an interpretable diagnosis system based on retrieving the most similar training sample (although it is unclear whether this actually helps physicians, as mentioned above), we can perform sample-wise representation learning, and use individual samples for retrieval. I do not see why retrieving a patient-level prototype (instead of a single ECG sample) provides any additional value regarding interpretability. Prototypes might be good for dataset distillation, but that is a completely independent topic.
>
> #### **Response to Comment 3**
> While it is possible to retrieve a single instance (as opposed to a PCP), doing so has several drawbacks. First, removing the PCP from the pipeline would make it difficult to reason at the patient level, which can be useful when probing the errors of a system (see Section 4.4). Second, PCPs can conceptually be extended to multiple data modalities, thereby succinctly summarizing the health state of a patient more broadly. For the implications of multi-modal prototypes, please refer to the Discussion. Moreover, from a computational standpoint, searching through the full dataset to retrieve a single instance is more demanding than searching through the PCPs, which can be orders of magnitude smaller in size. This can introduce unwanted latency during inference. We nonetheless conducted such experiments and found a minimal change in the diagnostic performance of the system.
>
> We have now included this Section 2.3 (page 3, last paragraph).

---

> > ### Author Response · Authors · 2022-11-25
> > **Response to Reviewer FnHS (Part 2)**
> >
> > ### **Comment 4**
> > The paper is overall very verbose. I see the same content repeatedly described multiple times throughout the paper. One example is how the paper repeatedly describe how they used short time-span ECG samples to ensure intra-patient invariance. The whole material of this paper does not seem to require 16 pages.
> >
> > #### **Response to Comment 4**
> > We have gone through the manuscript again from beginning to end, removing superfluous and redundant content. We have also relegated some content to the Appendices without interfering with the clarity of the manuscript. The main manuscript’s length has been reduced from 16 pages to 12 pages.
> >
> > ### **Comment 5**
> > The paper needs to rigorously address all weaknesses mentioned above. In my estimation, I see two possibilities:
> >
> > Carefully justify how retrieval-based diagnosis systems can help physicians in the real-world, when the physicians themselves don't even know the true label, and there is no way of knowing if the retrieved sample is indeed the most similar one to the given test sample, all the while the definition of "similar" is loosely defined. Then abandon prototypes because sample-wise retrieval can bring the same amount of interpretability as the prototype retrieval. Then demonstrate that sample-wise retrieval does not hurt diagnosis performance compared to none-retrieval-based models. This would naturally lead to abandoning all claims on dataset distillation.
> >
> > Withdraw any claims regarding interpretability. Stick to PCP, but only in the context of dataset distillation. But there can always be a more creative solution, and as long as the weaknesses are resolved, I would be happy to re-evaluate this work.
> >
> > #### **Response to Comment 5**
> > While we appreciate the reviewer’s two suggested paths here, we believe that there is indeed a third solution. We have removed claims regarding the interpretability of our framework and have instead more precisely stated the utility of PCPs in probing deep learning-based predictions. These changes are reflected in the framing of the paper, and predominantly in Section 4.4 and via a title change (it no longer states “interpretable medical diagnoses”).
> >
> > We also believe that removing our claim about interpretability does not necessarily mean we must discard our remaining results reflecting the utility of PCPs for other tasks (e.g., cardiac arrhythmia classification and patient retrieval). To address this, we have dedicated a paragraph to outlining the benefits of learning prototypes, at the patient level, compared to simply retrieving and working with instances (page 3, last paragraph).

---

> > > ### Comment · Reviewer_FnHS · 2022-12-21
> > > **Response to authors**
> > >
> > > Thank you for the detailed response, and the revised manuscript. I am mostly convinced by the arguments and the changes made to the manuscript, except for one minor issue.
> > >
> > > While the authors advocate the utility of their patient-level prototypes (to which I agree to some extent), it is only viable under their data processing constraint, where the authors "had to" use a single patient's ECG samples that are temporally close to one another (otherwise, the patient physiology might change over time, and his/her ECG patterns could no longer represent a single physiology). If this work was to be used over a longer timespan, the claims regarding the utility of patient-level prototypes would not stand. This is already addressed at the bottom of page 11, but it would make the paper more rigorous if the authors were able to address how PCP still has an advantage over sample-wise retrieval in the "longer timespan scenario", or how PCP can be easily modified to embrace "longer timespan scenario". In its current state, I'm not sure if readers would be convinced to use PCP when it clearly has a scalability issue.
> > >
> > > Other than this I think the paper seems to have gone through significant improvement.

---

### Review · Reviewer_NeHb · 2022-11-22

**Summary Of Contributions:**

The paper proposes a framework for diagnosing with ECG data. The framework leverages prototype embeddings that are jointly trained with the ECG representation. The paper also shows that the trained prototype can be used for interpreting the network, patient retrieving, and distillation.

**Audience:**

Yes

**Broader Impact Concerns:**

There is no ethical concern.

**Claims And Evidence:**

No

**Requested Changes:**

1. Compare with baselines such as CE loss only and unsupervised prototype-based contrastive learning methods
2. Provide more details about the tasks, baselines, and metrics.
3. Be cautious about claiming novelty.

**Strengths And Weaknesses:**

Strength:

- The paper explores several use cases for the trained prototypes in the application of healthcare.
- The prototypes are useful in the interpretation of the learned representations.

Weakness:

- Missing an important unsupervised prototype-based contrastive learning work [1]
- Where are the results of experiments mentioned in Section 3.3.2?
- The paper just skim through dataset distillation and patient retrieval. It's hard to understand what these tasks are, what other baselines do, and why these tasks matter.
- The claim of novelty in architecture is questionable since it looks quite similar to the attention mechanism. If we view the prototype as word embeddings, a similar architecture is widely used in NLP.
- It is unclear whether the proposed method can obtain a better classification performance than cross-entropy.

[1] Prototypical Contrastive Learning of Unsupervised Representations. Li et. al. 2020

---

> ### Author Response · Authors · 2022-11-25
> **Response to Reviewer NeHB (Part 1)**
>
> ### **Comment 1**
>
> Missing an important unsupervised prototype-based contrastive learning work [1]
> [1] Prototypical Contrastive Learning of Unsupervised Representations. Li et. al. 2020
>
> #### **Response to Comment 1**
> We have now made reference to this work, and outline how it is distinct from ours (Introduction, page 1, paragraph 3).
>
> ### **Comment 2**
> Where are the results of experiments mentioned in Section 3.3.2?
>
> #### **Response to Comment 2**
> To avoid any confusion about which ablation experiments correspond to which results, we have rewritten the Experimental design section (page 5) such that it directly links to the appropriate findings in the Results section.
>
> ### **Comment 3**
> The paper just skim through dataset distillation and patient retrieval. It's hard to understand what these tasks are, what other baselines do, and why these tasks matter.
>
> #### **Response to Comment 3**
> To gain a better appreciation of the tasks we have proposed (e.g., cardiac arrhythmia classification, dataset distillation, and patient retrieval), we have rewritten the Experimental design section (page 5), which now clearly delineates the three tasks, provides a description of each, outlines baseline strategies, and links to the corresponding results in the Results section. In light of Reviewer FnHS’s and 9duA’s comments about the verbosity of the first part of the manuscript (up until the results), we had to strike a balance between including a sufficient amount of details for understanding and not overwhelming the readers with too much information in one go. Additional details have been provided, where necessary, in the Appendix.
>
> As for understanding the importance of patient retrieval, we had outlined (Introduction, paragraph 2) a patient data retrieval system can be used by “physicians to compare the disease and treatment trajectories of similar-looking patients, and enable medical educators to leverage up-to-date real-world evidence as a means of teaching the next generation of medical students.”
>
> Furthermore, the implications of our findings, and by extension the tasks we have proposed, are outlined in the Discussion section (page 11, paragraph 1).
>
> ### **Comment 4**
> The claim of novelty in architecture is questionable since it looks quite similar to the attention mechanism. If we view the prototype as word embeddings, a similar architecture is widely used in NLP.
>
> #### **Response to Comment 4**
> While we do not believe that we had made claims about the novelty of the architecture, we have now double-checked the content of the manuscript to remove such claims. The reviewer is correct in that our approach, particularly during the inference stage, is akin to cross-attention in a Transformer architecture and even to edge connections in a graph. As we believe this analogy can improve the understanding of the content by readers, we have incorporated it into our manuscript (page 3, last paragraph) and into Figure 1 (page 4).
>
> ### **Comment 5**
> It is unclear whether the proposed method can obtain a better classification performance than cross-entropy.
>
> #### **Response to Comment 5**
> We had conducted experiments with only a supervised loss (i.e., cross-entropy) and these are now reflected in Figure 3 (right). While we did observe that a system learned without a contrastive loss (and without prototypes) performed marginally better than one learned with a contrastive loss (AUC 0.90 vs. 0.88). It is unknown whether this difference leads to meaningful and noticeable changes within the clinic. More broadly, however, it remains an open question whether there exists a trade-off between system transparency and performance, and in such an event, whether clinical stakeholders are willing to sacrifice some performance in exchange for the improved transparency and widened application areas of deep learning systems.
>
> We have included this in the Discussion section (page 12, paragraph 1).
>
> ### **Comment 6**
> Compare with baselines such as CE loss only and unsupervised prototype-based contrastive learning methods
>
> #### **Response to Comment 6**
> Please refer to our response to Comment 5 on the topic of supervised learning alone (i.e., a cross-entropy loss). As for an unsupervised contrastive learning approach, we had provided evidence that such an approach (which we referred to as NCE loss only and is akin to the suggested reference by Li et al., 2021) performs worse than one which optimizes both a supervised and contrastive loss (Section 4.3.3, and Fig. 8 in Appendix F). This is expected, particularly when the tasks that we had defined (cardiac arrhythmia classification, dataset distillation, etc.) are based on the cardiac arrhythmia disease class which was used for supervision.

---

> > ### Author Response · Authors · 2022-11-25
> > **Response to Reviewer NeHB (Part 2)**
> >
> > ### **Comment 7**
> > Provide more details about the tasks, baselines, and metrics.
> >
> > #### **Response to Comment 7**
> > Please refer to our response to Comment 3 above.
> >
> > ### **Comment 8**
> > Be cautious about claiming novelty.
> >
> > #### **Response to Comment 8**
> > While we do not believe that we had made claims about the novelty of the architecture, we have now double-checked the content of the manuscript to remove such claims.

---

### Author Response · Authors · 2022-11-25
**Modified Manuscript Uploaded**

We would like to thank the reviewers for taking the time and effort to review our manuscript and for providing us with valuable feedback. We address your comments individually below.

To improve clarity, our rebuttal takes the form of a "point-by-point response" which includes the original comment provided by the reviewers and our response to each comment.

Our manuscript has also been updated to reflect the changes suggested by the reviewers. These changes are highlighted in yellow for ease of review.

We look forward to an engaging discussion!

---

### Author Response · Authors · 2022-12-23
**Latest Manuscript Uploaded**

We would like to thank the reviewers for taking the time and effort to read our modified manuscript. We have now incorporated your latest comments into the manuscript (**highlighted in blue**).

To ensure that the reviewers who have yet to comment are still able to track the original changes, we have kept those **highlighted in yellow** (first round of updates).

### Changes to the manuscript

#### **Reviewer FnHS**
We have now included a paragraph in the Discussion section (page 12) which outlines how one could go about learning prototypes "over longer time-spans" (e.g., on the order of years).

#### **Reviewer 9duA**
We have now explicitly referenced the VICReg manuscript that you kindly directed us to (Discussion section, page 12) and mentioned that it could be an interesting exploration for future work.